# Globoside and the mucosal pH mediate parvovirus B19 entry through the epithelial barrier

**Corinne Suter[1], Minela Colakovic[1], Jan Bieri[1], Mitra Gultom[2], Ronald Dijkman[2], Carlos Ros[1]\***

**1** Department of Chemistry, Biochemistry and Pharmaceutical Sciences, University of Bern, Bern, Switzerland, **2** Institute for Infectious Diseases, University of Bern, Bern, Switzerland

\* carlos.ros@unibe.ch

**Data Availability Statement:** All relevant data are within the manuscript and its Supporting Information files

## Abstract

Parvovirus B19 (B19V) is transmitted primarily via the respiratory route, however, the mechanism involved remains unknown. B19V targets a restricted receptor expressed in erythroid progenitor cells in the bone marrow. However, B19V shifts the receptor under acidic conditions and targets the widely expressed globoside. The pH-dependent interaction with globoside may allow virus entry through the naturally acidic nasal mucosa. To test this hypothesis, MDCK II cells and well-differentiated human airway epithelial cell (hAEC) cultures were grown on porous membranes and used as models to study the interaction of B19V with the epithelial barrier. Globoside expression was detected in polarized MDCK II cells and the ciliated cell population of well-differentiated hAEC cultures. Under the acidic conditions of the nasal mucosa, virus attachment and transcytosis occurred without productive infection. Neither virus attachment nor transcytosis was observed under neutral pH conditions or in globoside knockout cells, demonstrating the concerted role of globoside and acidic pH in the transcellular transport of B19V. Globoside-dependent virus uptake involved VP2 and occurred by a clathrin-independent pathway that is cholesterol and dynamin-dependent. This study provides mechanistic insight into the transmission of B19V through the respiratory route and reveals novel vulnerability factors of the epithelial barrier to viruses.

## Author summary

The mechanism of entry of parvovirus B19 through the respiratory epithelium remains elusive. By using polarized MDCK II and well-differentiated primary human airway epithelial cell cultures, we revealed that the virus exploits the natural acidic environment of the nasal mucosa and the expression of globoside in the ciliated cell population of the airway to breach the respiratory epithelium by transcytosis. Mechanistically, virus entry and intracellular trafficking in the epithelial cells differ substantially from that in the permissive erythroid progenitor cells.

**Funding:** This study was supported by a grant from the Swiss National Science Foundation (grant 320030_207850 to C.S.). The funders had no role in study design, data collection and analysis, decision to publish, or preparation of the manuscript.

**Competing interests:** The authors have declared that no competing interests exist

## Introduction

Parvovirus B19 (B19V) is a highly prevalent human pathogen classified within the genus *Erythroparvovirus* of the family *Parvoviridae*. The single-stranded DNA genome encodes two structural proteins, VP1 and VP2, which assemble into a small (25 nm) non-enveloped icosahedral capsid [1]. The two structural proteins are identical except for an additional N-terminal extension in the VP1, the so-called VP1 unique region (VP1u), which harbors domains essential for the infection.

The infection in children is common and is typically associated with a rash disease named *erythema infectiosum* or fifth disease [2]. In adults, B19V can cause a wide range of syndromes, from mild to severe, depending on host factors that remain poorly understood. Acute infections may result in high-titer viremia, and when the infection occurs during pregnancy in seronegative mothers, the virus can invade the placenta and infect the developing fetus, causing potentially severe perinatal complications, such as fetal anemia, non-immune fetal hydrops, and fetal death [3,4].

To initiate the infection, the virus enters through the upper respiratory tract [5], and spreads via the blood circulation to the bone marrow, where it replicates in the erythroid precursor cells (EPCs). Productive infection of EPCs results in cell lysis, accounting for the hematologic disorders associated with the infection [2]. Cellular factors required for replication, transcription, and splicing are restricted to EPCs at the Epo-dependent differentiation stages [6–9]. Viruses depending on such strict intracellular conditions for replication must rely on highly selective cell targeting mechanisms to exclusively enter the few cell types where replication can occur. In this manner, the virus circumvents the largely predominant non-permissive cells, which would otherwise act as decoy targets and lead to inefficient viral infection and dissemination. Accordingly, B19V likely targets an erythroid-specific surface molecule as an entry receptor. Compared to other parvoviruses, the VP1u of B19V has an additional N-terminal region targeted by neutralizing antibodies, denoting its importance in the infection [10,11]. We have previously shown that this region harbors a receptor-binding domain (RBD) required for virus uptake and productive infection [12]. The RBD structure adopts a spatial configuration of three alpha helices [13], and binds a receptor whose expression is restricted to erythroid cells [14], coinciding with the homing of BFU-E cells to the erythroblastic islands in the bone marrow and subsequent differentiation to CFU-E, proerythroblasts, and early basophilic erythroblasts [15]. The tyrosine protein kinase receptor UFO (AXL) was found to interact specifically with the RBD, and to contribute to virus uptake [16].

Besides the erythroid receptor, there should be another receptor with a broader expression profile to enable virus entry through the respiratory epithelium. The glycosphingolipid globoside or P antigen has been historically recognized as the primary cellular receptor of B19V [17]. Individuals with a rare mutation in the globoside synthase gene, do not express globoside and have no serological evidence of a previous infection [18]. Attempts to validate the interaction of the virus with globoside were not conclusive. While the interaction was observed with globoside incorporated into supported lipid bilayers [19], no detectable binding was observed in a study using multiple, highly sensitive methods [20], suggesting that the interaction must require specific conditions. In a previous study, we demonstrated that globoside does not function as the cellular receptor required for viral entry into permissive EPCs. However, we found that globoside is essential after virus uptake for productive infection [21]. In a recent study, we further investigated the function of globoside in B19V infection and revealed that B19V does interact with globoside, but exclusively under acidic conditions. This feature allows the interaction of incoming viruses with globoside inside the acidic endosomes, where they converge

after virus uptake. In the absence of globoside, incoming particles internalize normally but are arrested inside the endosomes and the infection is blocked [22].

The pH-dependent affinity modulation between B19V and globoside directly affects the virus tropism and pathogenesis. Under neutral pH conditions, which are characteristic of most extracellular environments of the organism, binding the virus with the ubiquitously expressed glycosphingolipid is impossible. This strategy restrains the redirection of the virus to non-permissive cells, promoting the selective targeting of the virus to the permissive EPCs in the bone marrow where productive infection occurs. For example, globoside is particularly abundant on the membrane of the non-permissive erythrocytes [23,24]. However, erythrocytes do not play a significant role as decoy targets during B19V viremia because the neutral pH of the blood prohibits the interaction with the glycosphingolipid [22]. However, considering that the glycosphingolipid globoside is expressed in multiple cells and tissues, naturally occurring acidic niches in the body become potential targets for the virus. One of these acidic microenvironments is found in the nasal mucosal surface, which in healthy subjects has an average pH of 5.5–6.5 [25–27]. This pH spectrum corresponds to the optimal conditions required for B19V binding to globoside (pH < 6.4) [22]. Systemic infection resembling all aspects of a natural infection, including high-titer viremia, was achieved by intranasal inoculation of B19V in seronegative volunteers [28], confirming that the nasal epithelium is a site of virus entry. However, the viral and cellular factors that mediate the transmission of B19V through the respiratory epithelium remain unknown.

In this study, a genetically modified epithelial barrier and well-differentiated primary human airway epithelial cells were used as models to explore the mechanism of virus entry through the respiratory route. The results reveal that globoside is expressed in the human respiratory epithelium and, in concert with the natural acidic pH of the nasal mucosa, mediates virus transcytosis across the epithelial barrier without productive infection. Virus entry through the respiratory route involves a cellular receptor and follows a pathway that differs from those employed to target and productively infect the permissive erythroid progenitor cells in the bone marrow.

## Results

### B19V binding and internalization in MDCK II cells requires globoside and low pH

The Madin–Darby canine kidney cell line, MDCK II, is widely used as a model for studying the epithelium as they have clear apical-basolateral polarity with well-defined cell junctions [29]. Besides, the MDCK II cell line is readily amenable to targeted mutagenesis, which facilitates functional studies of cellular receptors.

Confocal immunofluorescence microscopy with a specific antibody revealed the expression of globoside in MDCK II cells (Fig 1A). Recombinant VP1u constructs with full-length or truncated receptor-binding domain (RBD) (S1A Fig) were used to examine the expression of the VP1u erythroid receptor [12,15]. The capacity of the functional VP1u construct to detect the expression of the VP1u receptor was evaluated in susceptible UT7/Epo cells (S1B Fig). In contrast to the abundant expression of globoside, the VP1u receptor was not detectable in MDCK II cells (Fig 1A).

Interaction of B19V with MDCK II cells was exclusively detected under low pH conditions. Cells appeared with a dense perinuclear signal, characteristic of the accumulation of incoming particles in the endocytic compartment (Fig 1B). The pH-dependent interaction of B19V with MDCK II cells was further confirmed and quantified by qPCR (Fig 1C).

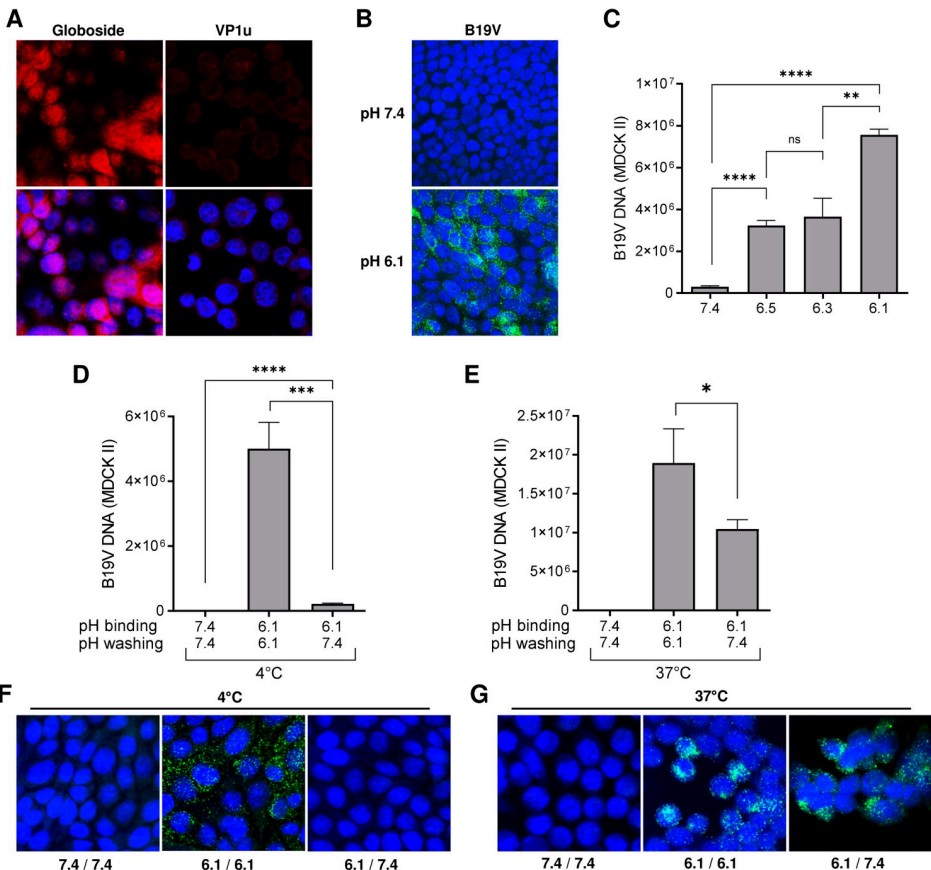

**Fig 1. MDCK II cells express globoside and support B19V attachment and uptake under acidic conditions.** (A) Detection of globoside and VP1u erythroid receptor in MDCK II cells by IF. Globoside expression was examined with a specific antibody followed by a secondary Alexa Fluor 594 antibody. VP1u receptor expression was examined with a recombinant VP1u construct, a rat anti-FLAG antibody, and a secondary Alexa Fluor 594. Nuclei were stained with DAPI (B) Detection of B19V capsids (860-55D) in MDCK II cells under neutral or acidic pH by IF. (C) Quantitative analysis of B19V binding to MDCK II cells under different pH conditions. B19V was incubated with cells at different pH conditions at 4˚C. After 1h, cells were washed in their respective buffers, DNA was extracted, and viral DNA quantified by PCR. B19V was incubated with cells at 4˚C and pH 7.4 or 6.1. After 1h, cells were washed with PiBS (pH 6.1) or PBS (pH 7.4) (D and F) or further incubated at 37˚C for 1h before washing (E and G). Subsequently, B19V DNA was detected by quantitative PCR (D and E) or IF with an antibody against intact capsids (F and G). The results are presented as the mean ± SD of three independent experiments.

In previous studies, we showed that the binding of B19V to globoside is reversible and the complex could be dissociated when exposed to neutral pH [22]. The virus bound to MDCK II cells at 4˚C was efficiently removed after washing the cells with PBS (pH 7.4) (Fig 1D). In contrast, when cells were incubated at 37˚C to allow virus uptake, a large proportion of viral particles (55.2% ±6) could not be removed at neutral pH, indicating that they were internalized (Fig 1E). Immunofluorescence microscopy further confirmed this result (Fig 1F and 1G).

To ascertain the role of globoside in the interaction of B19V with MDCK II cells, the B3GalNT1 gene coding for globoside synthase was knocked out in MDCK II cells (S2 Fig). The loss of this enzyme, which catalyzes the conversion of globotriaosylceramide to globoside, leads to the elimination of globoside and all downstream glycosphingolipids. The presence of B3GalNT1 mRNA was examined in the wild-type (WT) and transfected cells by RT-PCR. While an amplicon of the expected size was detected in WT cells, no detectable signal was

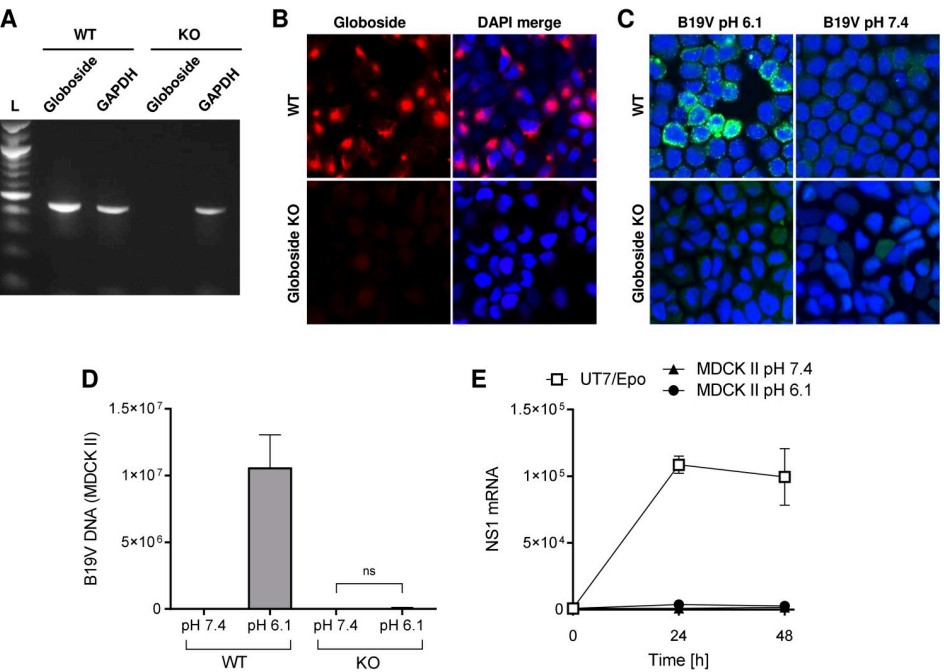

**Fig 2. Globoside and acidic pH mediate B19V uptake in MDCK II cells.** (A) The B3GalNT1 gene, coding for globoside synthase, was knocked out in MDCK II cells. Total poly(A) mRNA was isolated and used to detect B3GalNT1 mRNA by RT-PCR in the wild-type (WT) and globoside knockout cells (KO). Amplicons were analyzed by agarose gel electrophoresis. Detection of GAPDH RNA was used as a loading control. L, DNA ladder. (B) The expression of globoside was examined in WT and KO cells by IF with a specific antibody. (C) B19V was incubated with WT or KO cells for 1h at 37˚C at pH 6.1 or 7.4. Subsequently, cells were washed with PBS (pH 7.4) to remove non-internalized virus. B19V capsids (860-55D) were detected by IF. (D) B19V was incubated with WT or KO cells for 1h at 37˚C at pH 6.1 or 7.4. Subsequently, cells were washed with PBS (pH 7.4) to remove non-internalized viruses. Total DNA was extracted, and the amount of viral DNA was quantified by qPCR. (E) Detection of NS1 mRNA in MDCK II cells infected with B19V under neutral and acidic conditions. After 1h, cells were washed or further incubated for 24h and 48h at 37˚C. NS1 mRNA was extracted and quantified by RT-qPCR. UT7/Epo cells were used as a positive control. The results are presented as the mean ± SD of three independent experiments. ns, not significant.

observed in the transfected cells (Fig 2A). The knockout of B3GalNT1, was confirmed by immunofluorescence microscopy using a specific antibody against globoside [30] (Fig 2B).

WT and globoside knockout MDCK II cells (KO) were used to compare B19V binding and internalization by confocal immunofluorescence microscopy. B19V was incubated with the cells for 1h at 37˚C. After several washing steps, the cells were fixed and stained with antibody 860-55D against intact capsids. As expected, B19V was able to interact with WT cells at acidic pH (6.1) but not at neutral pH. In globoside KO cells, no signal was detectable at any pH conditions (Fig 2C). The capacity of B19V to bind and internalize into WT and globoside KO cells was examined by quantitative PCR (qPCR). While B19V DNA was detected in cells expressing globoside at acidic pH, no significant signal was detected in globoside KO cells at neutral or acidic pH (Fig 2D).

B19V has a narrow tissue tropism and requires specific intracellular factors expressed exclusively in human erythroid precursor cells for infection [2]. MDCK II are non-erythroid canine cells and despite the presence of intracellular B19V, they do not provide the required intracellular environment to support B19V infection. As expected, NS1 mRNA was not detected in infected MDCK II cells at any pH condition (Fig 2E).

## Transcytosis of infectious particles is mediated by globoside and acidic pH

The capacity of B19V to cross the MDCK II monolayer barrier and reach the basolateral side was evaluated. To this end, MDCK II cell monolayers were grown to confluence on tissue culture (TC) inserts (Sarstedt, Newton, NC) with 0.4 μm pores to constitute a biological barrier. The functionality of the tight junctions and the formation of a selective cellular barrier were assessed at increasing days post-seeding at pH 7.4, where no interaction of B19V with globoside is possible, and only passive diffusion allows the transfer of particles to the basolateral side. At 24h post-seeding, B19V was detected at the basolateral side, denoting the lack of functional tight junctions. At 48 and 72h post-seeding, B19V was not detectable on the basolateral side, indicating the establishment of functional tight junctions (S3A Fig), as shown by confocal microscopy (S3B Fig). Transcytosis assays were performed on day 3 post-seeding. B19V was added to the apical side at neutral (7.4) or at acidic pH (6.1) and the presence of viruses in the basolateral medium was examined at increasing times post-inoculation by qPCR. The results showed that B19V was able to access the basolateral side in a time-dependent manner but exclusively when cells were exposed to acidic conditions at the apical side (Fig 3A). Similar results were obtained using membranes with different pore densities and thicknesses (S4 Fig). Confocal Z-stack sectioning from the apical to the basal side revealed the presence of internalized viral particles distributed throughout the entire cellular barrier (S5 Fig).

The rate of apical-to-basal transport was quantitatively analyzed by comparing the viral particles in the apical medium (input) with the viral particles accumulated in the basolateral medium following incubation at 37°C. Approximately 8.1% (±2.12) of viruses crossed the polarized cell monolayer barrier within 3h (Fig 3B).

To ascertain the role of globoside in transporting viral particles across the epithelial barrier, transcytosis assays were performed in WT and globoside KO cells at acidic pH. In cells expressing globoside, B19V accumulated in the basolateral side over time. In sharp contrast, transcytosis was not detected in globoside KO MDCK II cells (Fig 3C).

To confirm that the viral DNA detected in the basolateral medium originates from intact virions and not from free viral DNA or DNA associated with lipids, the basolateral medium was treated with micrococcal nuclease alone or in combination with NP40 before qPCR. These treatments did not affect the amount of viral DNA detected in the medium by qPCR, confirming that the detected DNA originates from intact virions (S6 Fig). Moreover, viruses recovered in the basal medium following transcytosis through MDCK II cells remained infectious, as demonstrated by their capacity to infect UT7/Epo cells (Fig 3D).

## Tight junctions remain functional during B19V transcytosis

Viruses may interact with tight junction elements compromising the integrity of the cellular barrier and promoting paracellular transport of particles across the epithelium [31]. To verify a possible effect of low pH or B19V on the cell barrier integrity, the expression and intracellular distribution of ZO-1 and claudin-1, two integral membrane proteins essential for the formation and functionality of tight junctions [32], were investigated. Polarized MDCK II cells were grown on TC inserts for 3 days to allow the formation of tight junctions (S3 Fig). The cells were infected with B19V for 24h or left uninfected, and the pericellular distribution of ZO-1 and claudin-1 was tested by immunofluorescence with specific antibodies. No significant difference in pericellular distribution and fluorescent intensity was observed between infected and non-infected cells (Fig 4A). Likewise, the expression profile of the tight junction proteins was unaffected after the incubation of cells with the virus for 24h (Fig 4B).

Although B19V does not appear to modify the intracellular distribution and expression levels of ZO-1 and claudin-1, we tested the functionality of the tight junctions by using PP7

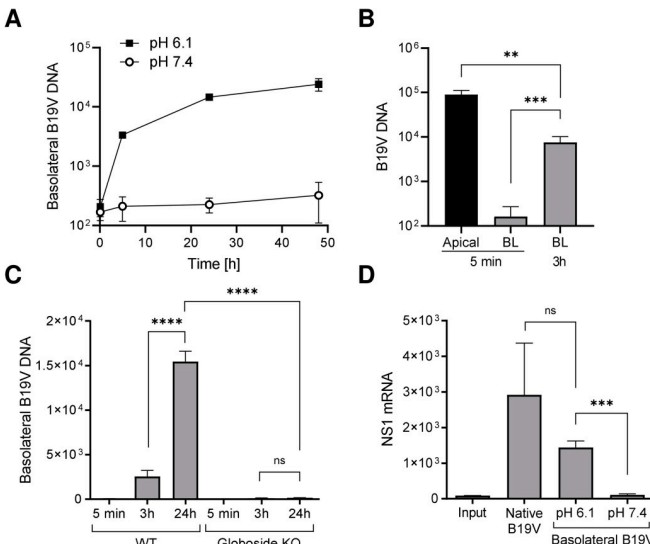

**Fig 3. Transcytosis of infectious particles in MDCK II cells.** (A) MDCK II cell monolayers were grown to confluence on TC inserts. B19V ($3\times10^9$) was added to the apical side at neutral or acidic pH and incubated at 37°C. At increasing times post-inoculation, DNA was extracted from a fraction of the basolateral medium and viral DNA was quantified by qPCR. (B) The rate of apical-to-basal transport was quantitatively analyzed by comparing the viral particles in the apical side with the particles accumulated in the basolateral medium. B19V ($3\times10^9$) was added to the apical side at acidic pH (6.1) and incubated at 37°C. At increasing times post-inoculation, viral DNA in the basolateral medium was quantified by qPCR and compared with the input virus in the apical medium. (C) Transcytosis assay in WT and globoside KO cells at acidic pH. B19V ($3\times10^9$) was added to the apical side and incubated at 37°C. At increasing times post-inoculation, DNA was extracted from a fraction of the basolateral medium, and viral DNA was quantified by qPCR. (D) Detection of NS1 mRNA in UT7/Epo cells infected with native B19V or with B19V present in the basolateral medium following a transcytosis assay for 24h at pH 6.1 or 7.4. Cells were infected with the same number of viral particles and NS1 mRNA was quantified by RT-qPCR. The results are presented as the mean ± SD of three independent experiments. ns, not significant.

bacteriophage. PP7 has a similar size as B19V, does not bind to eukaryotic cells, and contains an RNA genome that can be used for quantification [33]. PP7 particles were added to the apical medium at pH 7.4 or 6.1 in the absence or presence of B19V. Phage particles did not accumulate in the basolateral medium, confirming that tight junctions remain functional, and the cellular barrier maintains its integrity (Fig 4C and 4D).

## VP1 is dispensable for virus uptake in epithelial cells

VP1 plays a central role in B19V entry and intercellular trafficking. The N-terminal region of VP1 harbors a receptor-binding domain (RBD) and a phospholipase $A_2$ domain that are essential for virus uptake and endosomal escape, respectively [13,34]. To evaluate the role of the VP1 domains in globoside-mediated virus transcytosis through the epithelium, recombinant VP2 VLPs devoid of VP1 were produced in insect cells (S7 Fig). MDCK II cells were incubated under neutral or acidic pH conditions with VP2 VLPs at 37°C, washed at neutral pH to remove bound but not internalized capsids, and examined by immunofluorescence microscopy with an antibody against intact capsids. As shown in Fig 5A, VP2 VLP uptake was detected in cells incubated at 37°C under acidic pH and washed with PBS (pH 7.4) to remove bound but non-internalized particles. VLPs were not detected in cells incubated at neutral pH or in globoside KO cells at acidic conditions. Confocal Z-stack sectioning of MDCK II cells grown to confluence on TC inserts and incubated with VLPs revealed the presence of

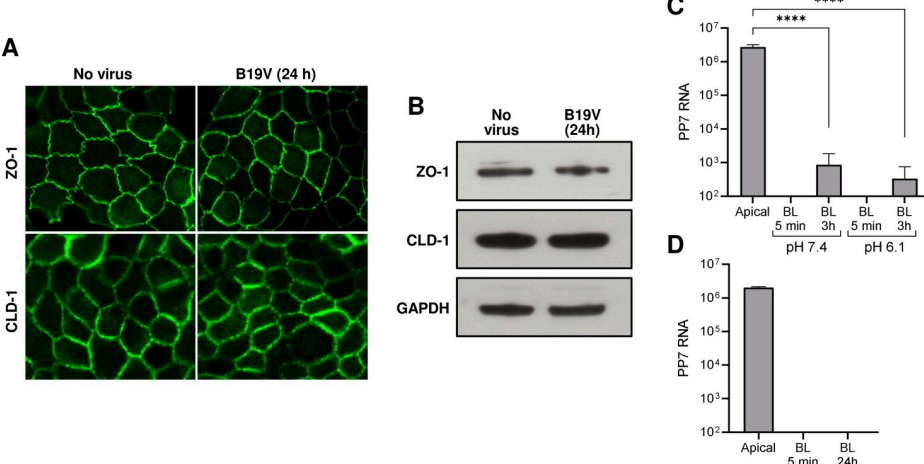

**Fig 4. Tight junctions remain functional during B19V transcytosis.** Expression and intracellular distribution of ZO-1 and claudin-1 in the presence or absence of B19V. MDCK II cells were grown on TC inserts with 0.4 μm pores for 3 days to allow the formation of tight junctions. The cells were infected with B19V for 24h or left uninfected and the pericellular distribution of ZO-1 and claudin-1 was tested by confocal immunofluorescence microscopy (A) or Western blot (B) with specific antibodies. GAPDH was used as a loading control. (C) The effect of low pH on the functionality of the tight junctions was tested by using PP7 bacteriophage. PP7 particles ($3\times10^{10}$) were added to the apical medium at pH 7.4 or 6.1. At 5 min and 24h, PP7 RNA in the basolateral medium was extracted and analyzed by quantitative RT-PCR and compared with the input in the apical medium. (D) The effect of B19V on the functionality of the tight junctions was tested by using PP7 bacteriophage. B19V ($3\times10^{9}$) and PP7 ($3\times10^{10}$) were added to the apical medium at pH 6.1. At 5 min and 24h, PP7 RNA in the basolateral medium was extracted and analyzed by quantitative RT-PCR and compared with the input in the apical medium. BL, basolateral. CLD-1, claudin 1. ZO-1, zonula occludens 1. The results are presented as the mean ± SD of three independent experiments.

internalized particles distributed throughout the entire cellular barrier (S5 Fig). These results indicate that VP1 is dispensable for B19V attachment and uptake in epithelial cells.

To further investigate the role of VP1, two neutralizing mAbs, one targeting the VP2 at the 5-fold region (860-55D) and another targeting the N-terminal of VP1u (1418) [35], were tested for their capacity to interfere with virus uptake and transcytosis. Antibody 1418 inhibits virus uptake in erythroid cells [12]. The antibody against VP2 hindered the interaction of the virus with globoside. In contrast, the antibody against VP1u did not significantly affect virus attachment (Fig 5B). To examine virus uptake in presence of the antibodies, viruses were incubated with cells at 4˚C and pH 6.1 to allow binding followed by incubation at 37˚C to allow virus internalization. Cells were subsequently washed at pH 7.4 to remove non-internalized viruses. Samples maintained at 4˚C served as a control. Virus uptake was inhibited by the VP2 antibody but not by the VP1u antibody (Fig 5C and 5D). The proportion of attached viruses able to internalize in the presence of the VP1u antibody was similar to that observed in the absence of antibodies (Fig 1E). We next tested the capacity of the antibodies to interfere with transcytosis. The accumulation of viral particles in the basolateral medium was blocked with the antibody against VP2 and substantially reduced with the antibody against VP1u (Fig 5E). These results confirm that VP1 is dispensable for virus internalization in epithelial cells; however, it facilitates transcytosis.

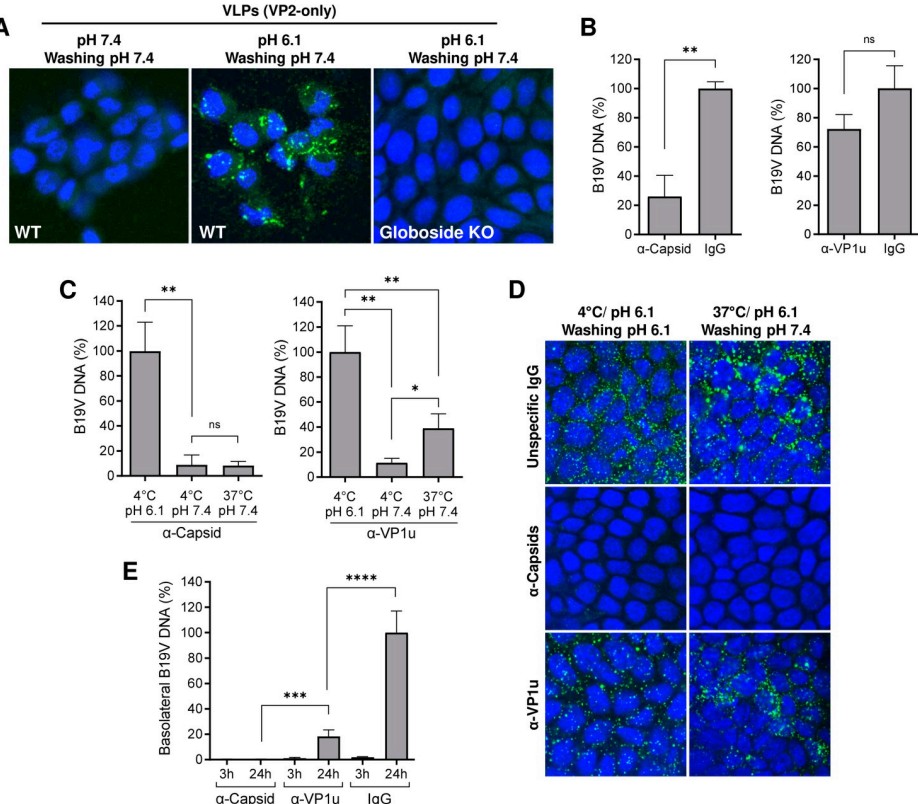

**Fig 5. VP1 is dispensable for virus uptake in epithelial cells.** (A) WT and globoside KO MDCK II cells (3x10⁵) were incubated with VP2 VLPs (3x10⁹) at 37˚C at neutral or acidic pH for 1h, washed at neutral pH to remove bound but not internalized capsids, and examined by IF with an antibody against intact capsids. (B) B19V attachment to MDCK II cells was quantified in the presence of antibodies against VP2 (α-Capsid; 860-55D; 0.3 μg) or VP1u (1418; 0.3 μg). An unspecific IgG preparation (human IgGs; 0.3 μg) was used as a control. The antibodies were incubated with the virus for 1h at 4˚C and added to the cells at pH 6.1 for 1h at 4˚C. Cells were washed at pH 6.1 and the viral DNA was quantified by qPCR. Alternatively, cells were washed at pH 6.1 or 7.4, to detached bound but not internalized capsids. Subsequently, B19V DNA was quantified by qPCR (C), and capsids were detected by confocal immunofluorescence microscopy with an antibody against intact capsids (D). (E) Virus transcytosis in MDCK II cells was quantified in the presence of antibodies against VP2 (860-55D; 0.3 μg) or VP1u (1418; 0.3 μg). An unspecific IgG preparation (human IgGs; 0.3 μg) was used as a control. The antibodies were incubated with B19V (3x10⁹) for 1h at 4˚C, added to the apical side at pH 6.1, and incubated at 37˚C. At the indicated time points, DNA was extracted from a fraction of the basolateral medium and viral DNA was quantified by qPCR. The results are presented as the mean ± SD of three independent experiments. ns, not significant.

## Differences in the pH control virus uptake at the apical side and release at the basolateral side

The pH-dependent interaction of B19V with globoside at the apical side allows virus attachment and transport across the cell to the basolateral side. We hypothesize that the pH dependence of binding maintains the virus bound to globoside in the acidic endosomes until the complex is transported to the basolateral side, where the neutral pH destabilizes the interaction, and the virus is released. To assess whether the neutral pH at the basolateral side is required to release viral particles, we lowered the pH of both the apical and basolateral medium from 7.4 to 6.1. As expected, acidification of the basolateral medium blocked the release of viral particles (Fig 6A). Immunofluorescence microscopy revealed the accumulation of capsids

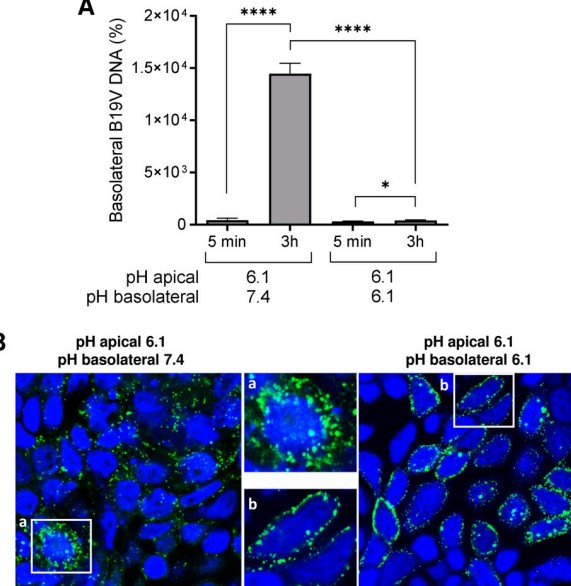

**Fig 6. pH controls the release of B19V at the basolateral side.** (A) Release of B19V at the basolateral side was tested under neutral and acidic conditions. B19V was added to the apical side at acidic conditions to allow virus entry and transcytosis and the basolateral pH was adjusted to 7.4 or 6.1. After 5 min (background control) and 3h, virus release in the basolateral medium was quantified by qPCR. (B) The intracellular distribution of B19V in MDCK II cells was examined by IF with an antibody against intact capsids under conditions blocking (pH 6.1) or allowing virus release (pH 7.4) at the basolateral side. AP, apical. BL, basolateral. The results are presented as the mean ± SD of three independent experiments. ns, not significant.

at a pericellular location (Fig 6B). These results demonstrate that the pH difference between the apical and basolateral compartments of the epithelium allows virus uptake and release.

## Virus uptake in epithelial cells occurs by a clathrin-independent pathway and is cholesterol and dynamin-dependent

In previous studies, we found that B19V uptake in erythroid progenitor cells is mediated by the interaction of VP1u with an erythroid receptor [12,15], and occurs via clathrin-mediated endocytosis [36]. To study the mechanism of B19V uptake in epithelial cells, MDCK II cells were treated with drugs interfering with clathrin and caveolae-mediated endocytosis. Pitstop 2 and chlorpromazine (CPZ) selectively block clathrin-mediated endocytosis. Pitstop blocks clathrin-coated pit dynamics and endocytosis [37], and CPZ prevents AP2-mediated assembly and disassembly of the clathrin lattice [38]. Genistein and methyl-β-cyclodextrin (mβCDX) inhibit caveolae-dependent endocytosis. mβCDX inhibits caveolae by cholesterol depletion [39], while genistein blocks caveolae-mediated internalization by inhibiting protein tyrosine kinases [40]. Polarized MDCK II cells grown on TC inserts were pre-treated with different doses of the drugs and infected with B19V. At 3h post-infection, cells were washed with PBS (pH 7.4) to remove bound but not internalized viruses. The cells were fixed, and internalized viruses were examined by confocal immunofluorescence. In cells where caveolar-like endocytosis was disturbed, virus internalization was inhibited. In contrast, drugs disturbing clathrin had no significant effect (Fig 7A). To further confirm these results, the accumulation of viruses in the basolateral compartment was quantified by qPCR. mβCDX and genistein showed a dose dependent decrease in viral transcytosis, in contrast, pitstop 2 and CPZ had no significant

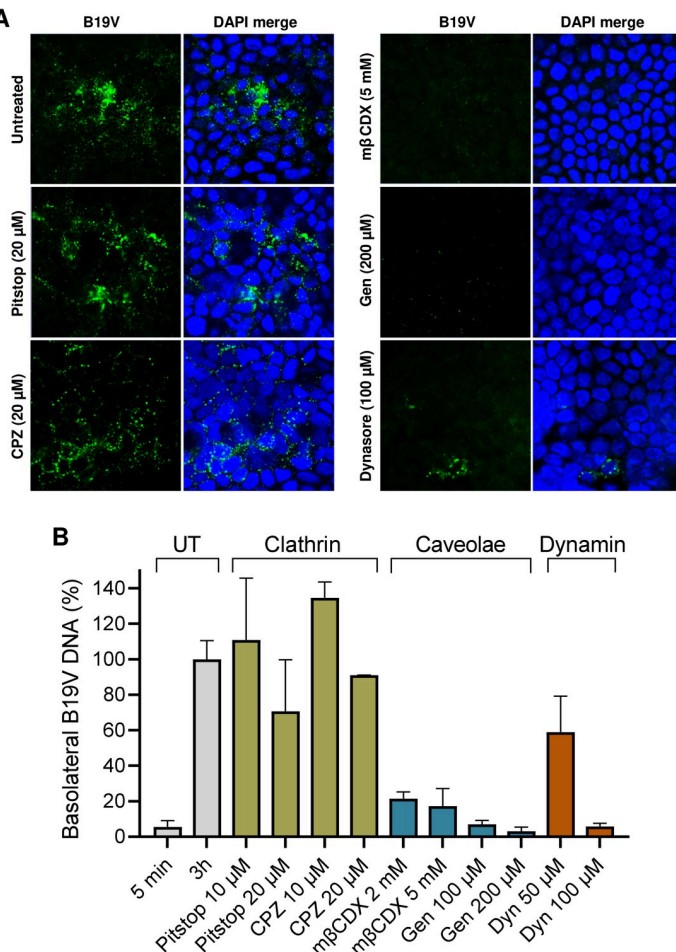

**Fig 7. Clathrin-independent, cholesterol and dynamin-dependent uptake of B19V in MDCK II cells.** MDCK II cells grown on TC inserts were pre-treated for 30 min with drugs interfering with clathrin, caveolae-mediated endocytosis, and dynamin function, and infected with B19V at pH 6.1. (A) At 3h post-infection, cells were washed with PBS (pH 7.4) to remove bound but not internalized virus. The cells were fixed, and internalized viruses were examined by IF with an antibody against intact capsids. (B) The accumulation of viruses in the basolateral medium was quantified by qPCR. The untreated sample (3h) was set to 100%. UT, untreated. CPZ, chlorpromazine. mβCDX, methyl-β-cyclodextrin. Gen, genistein. Dyn, dynasore. The results are presented as the mean ± SD of three independent experiments.

effect (Fig 7B). These results indicate that globoside-mediated B19V entry in epithelial cells occurs via a clathrin-independent, caveolar-like pathway.

Clathrin-independent endocytosis can be dynamin-dependent and dynamin-independent [41]. The ubiquitously expressed GTPase dynamin is essential for the formation of clathrin-coated vesicles as well as for ligand uptake through caveolae [42]. Dynasore is a potent inhibitor of dynamin-dependent endocytic pathways by blocking vesicle formation [43]. Treatment of MDCK II cells with dynasore resulted in a significant reduction of B19V uptake and transcytosis (Fig 7A and 7B).

Taken together, these results indicate that the viral and cellular factors involved in B19V entry and transcellular trafficking through the epithelium are different from those involved in virus uptake and productive infection in erythroid progenitor cells.

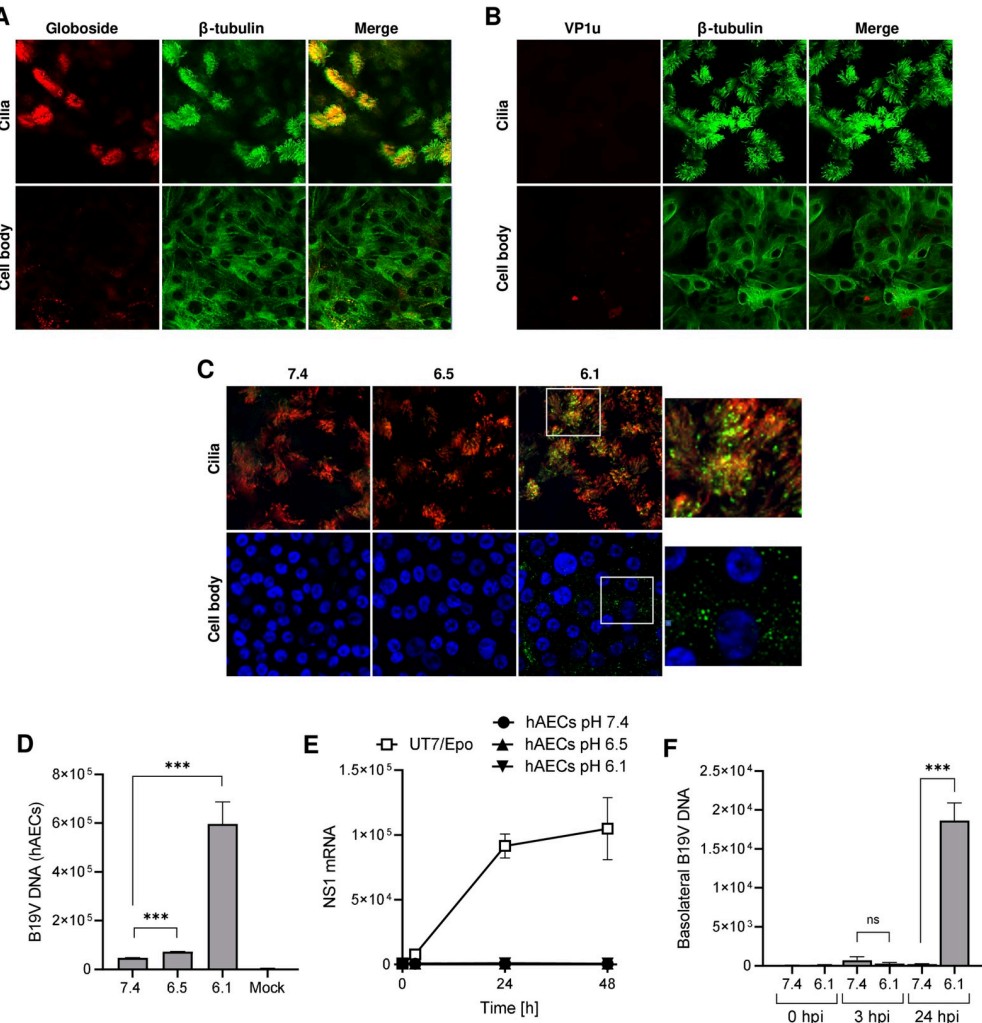

**Fig 8. Primary human airway cells express globoside and support B19V uptake and transcytosis under acidic conditions.** (A and B) Detection of globoside and VP1u receptor in hAEC cultures by IF. Globoside expression was examined with a specific antibody followed by a secondary Alexa Fluor 594 antibody. VP1u receptor expression was examined with a recombinant VP1u construct and an anti-FLAG antibody. An antibody against β-tubulin was used to label cilia and DAPI was used to label nuclei. Images were taken at the intersection level of the cilia and cell body. (C) Detection of B19V in hAECs incubated at different pH conditions by IF with antibody 860-55D against intact capsids, and a secondary Alexa Fluor 488 antibody. DAPI and β-tubulin were used to detect nuclei and cilia, respectively. (D) Detection of B19V in hAECs incubated at different pH conditions by quantitative PCR. (E) Detection of NS1 mRNA in hAECs infected with B19V under neutral and acidic conditions. Cells were washed after 1h or further incubated for 24h and 48h at 37˚C. NS1 mRNA was extracted and quantified by RT-qPCR. UT7/Epo cells were used as a positive control. (F) Transcytosis assay in hAECs. B19V was added to the apical side of the cells at neutral (7.4) or acidic pH (6.1), and the presence of viruses in the basolateral medium was examined by qPCR. The results are presented as the mean ± SD of three independent experiments.

## Primary human airway epithelial cell cultures express globoside and support virus transcytosis under acidic conditions

Although the MDCK cell line is a well-established epithelial model, it does not recapitulate the complexity of the respiratory epithelium. Cultured well-differentiated human airway epithelial cell (hAEC) cultures mimic the human airway microenvironment and facilitate the study of virus interactions with the respiratory epithelium. Isolation and culturing of hAEC cultures

were performed as previously described and resulted in a typical pseudostratified appearance [44] (S8 Fig).

Confocal immunofluorescence microscopy with a specific antibody revealed the selective expression of globoside in the ciliated population of the airway, particularly abundant at the middle and proximal domains of the cilia (Fig 8A). In contrast, the VP1u erythroid receptor was not detectable in any cell type of the airway epithelium (Fig 8B).

Virus attachment and uptake were examined under different pH conditions by confocal immunofluorescence microscopy using an antibody against intact capsids (860-55D) at 37˚C for 1h. As shown in Fig 8C, B19V was detected in the cilia and the cell body exclusively under acidic conditions (pH 6.1). Bound but not internalized viral particles were removed by exposing the cells to neutral pH, which has already been shown to trigger the dissociation of the virus from globoside [22], and the remaining virions were quantified by qPCR. As expected, virus internalization was not detected under neutral conditions and was strongly stimulated under acidic conditions (Fig 8D).

The capacity of the internalized B19V to productively infect hAEC cultures was analyzed. hAECs were incubated with B19V at neutral or acidic pH to allow virus uptake. At progressive times post-infection, the presence of NS1 mRNA was quantitatively determined by RT-PCR. UT7/Epo cells were used as a positive control. NS1 mRNA was not detected in hAECs at any pH conditions or time post-infection (Fig 8E), confirming that the internalized virions do not initiate a productive infection.

A transcytosis assay was performed to verify the capacity of B19V to cross the airway epithelium and reach the basolateral side. To this end, hAEC cultures were grown on Transwell inserts. The functionality of the tight junctions and the development of a selective cellular barrier were tested by the transepithelial electrical resistance (TEER) (S9 Fig). B19V was added to the apical side of the cells at neutral (7.4) or acidic pH (6.1), and the presence of viruses at the basolateral side was examined by qPCR. The results showed that B19V could access the basolateral side exclusively when cells were exposed to the pH conditions that allow virus uptake (pH 6.1) (Fig 8F).

In summary, the ciliated cells of the human airway express globoside and support B19V uptake and transcytosis without productive infection. The transport of viral particles across the airway epithelium was detected exclusively under the acidic conditions characteristic of the nasal mucosa [25–27], which correspond to the pH required for the interaction of the virus with soluble or membrane-associated globoside [22]. The VP1u erythroid receptor responsible for B19V internalization in EPCs was not detectable in the hAECs, excluding its role in virus uptake in the airway epithelium.

## Discussion

The epithelial cells lining the respiratory tract provide a tight barrier against pathogens. However, viruses have evolved strategies to breach the respiratory epithelium by taking advantage of the available cellular factors and environmental conditions. B19V is a highly prevalent human pathogen transmitted primarily through the respiratory route by aerosol droplets [5]. Systemic infection was achieved by intranasal inoculation in seronegative volunteers [28], indicating that viral particles interact with the upper respiratory epithelium to initiate the infection. However, the viral and host factors mediating the entry of B19V through the epithelium have yet to be discovered.

B19V requires strict intracellular conditions for replication, which can only be found in erythroid progenitor cells (EPCs) in the bone marrow. Consequently, B19V must rely on a highly selective cell targeting mechanism to efficiently traffic from the site of entry in the respiratory

epithelium to the distant site of replication in the bone marrow. A receptor expressed exclusively in the target erythroid progenitor cells would facilitate the circulation of the virus through non-permissive cells and tissues. In line with this strategy, we found that the VP1u of B19V harbors an RBD, which interacts with an erythroid-specific surface molecule(s) required for virus internalization in the permissive EPCs [12,14,15]. Recently, the tyrosine protein kinase receptor UFO (AXL) was found to interact with the VP1u and to be required for virus uptake [16]. In the airway, AXL is expressed exclusively in the basal cells, which are progenitors of all cells in the respiratory epithelium. Basal cells reside in a deep layer of the epithelium anchored to the basal lamina, where they are isolated from the external environment [45]. None of the exposed cells of the airway express receptor(s) interacting with VP1u, accordingly, B19V must use another receptor(s) to enter through the airway. The glycosphingolipid globoside is essential for the infection [21]. In an earlier study, we found that B19V binds globoside exclusively under acidic conditions [22]. Considering that globoside is expressed in multiple tissues, those exposed to acidic conditions, such as the nasal mucosa (average pH of 5.5–6.5 in healthy subjects) [25–27], become potential targets for the virus.

In this study, we used the MDCK II cell line and hAECs as models to study the interaction and transport of B19V across the epithelium. MDCK II cells express globoside and support B19V uptake and transcytosis exclusively under acidic conditions. In sharp contrast, virus attachment and subsequent internalization and transcytosis were abolished in globoside knockout cells regardless of the pH conditions, confirming the concerted role of globoside and pH in the transcellular transport of the virus across the epithelium. The VP2 mediates the binding of B19V to globoside and subsequent uptake, as VLPs devoid of VP1 can internalize in MDCK II under acidic conditions but not in globoside knockout cells. Binding and uptake were not disturbed by an antibody against VP1u, however, transcytosis was significantly reduced, suggesting that following internalization, VP1u facilitates the transcellular trafficking of B19V.

Well-differentiated airway epithelial cell cultures grown on porous membranes recapitulate the complex microenvironment of the human airway and are helpful to study the interaction of viruses with the human respiratory tract. We found that the ciliated cells of the upper respiratory epithelium express globoside and support virus uptake and transcytosis in concert with the natural acidic pH of the nasal mucosa. Virus uptake by airway epithelial cells did not result in productive infection. This was expected, as the VP1u receptor(s) and the strict intracellular conditions required for B19V replication, such as hypoxia [46,47], and Epo signaling [9], are absent in the upper airway epithelium.

Globoside is typically found in lipid rafts, which are membrane microdomains enriched in cholesterol and glycosphingolipids [48]. Accordingly, drugs interfering with cholesterol and caveolar function blocked virus transcytosis. Membrane glycosphingolipids are also exploited by other pathogens to breach the epithelium by transcytosis. Endocytosis of viruses, such as simian virus 40 [49], and norovirus [50], is mediated by multimeric binding to membrane glycosphingolipids resulting in membrane invagination and vesicle formation. A similar mechanism, involving glycosphingolipids and membrane deformation, is involved in the transcellular transport of Shiga and cholera toxins [51,52]. Whether similar changes in membrane properties are induced by the multimeric interaction of B19V with globoside will need further investigation. Mechanistically, B19V entry in epithelial cells differs substantially from permissive EPCs. In EPCs, virus uptake is mediated by the interaction of the receptor-binding domain in the VP1u with an erythroid receptor [15], followed by clathrin-mediated endocytosis [36]. Uptake in epithelial cells is mediated by the interaction of the VP2 with globoside, followed by internalization via a clathrin-independent, caveolae-like pathway.

The pH-mediated modulation of receptor affinity enables virus binding to globoside and uptake at the acidic apical surface of the airway and dissociation at the neutral basolateral side. This switching mechanism allows the virus to circulate free in the neutral pH environment of the blood without being trapped by globoside expressed in the non-permissive red blood cells, promoting efficient virus dissemination to the bone marrow.

Individuals with the rare phenotypes P(1)(k) and P(2)(k) have a mutation in the globoside synthase gene and cannot synthesize globoside [53]. These individuals are not susceptible to B19V infection and do not have serologic evidence of previous infections [18]. Our study supports this observation since in the absence of globoside, the virus cannot breach the airway epithelium and get exposed to the immune system.

B19V can spread transplacentally to the developing fetus [54,55]. Globoside expression has been detected in the villous trophoblast during the first trimester of pregnancy, and the expression decreases progressively during gestation [56]. The progressive decrease of globoside expression relative to gestational age correlates with the higher risk of fetal infection [3]. The acidic pH required for the interaction may be generated locally by sodium (Na+)/H+ exchangers (NHEs) expressed in the vascular endothelium and syncytiotrophoblast [57,58]. A detailed characterization of the level and dynamics of globoside expression, cellular distribution, and microenvironmental conditions of the placenta will be required to evaluate the role of globoside as a risk factor for fetal infection.

In summary, this study reveals the mechanism of parvovirus B19 entry through the epithelial barrier. In concert with local acidic conditions, the virus exploits the natural expression of globoside to breach the airway epithelium by transcytosis. To enter epithelial and erythroid cells, B19V exploits distinct receptors, capsid domains, and endocytic pathways, resulting in transcellular transport and productive infection, respectively. The detailed atomic structure of the binding site of globoside and the pH-mediated capsid rearrangement that controls the affinity for the receptor will be essential to assist the rational design of small interfering molecules and specific antibodies aiming to prevent B19V from breaching host cellular barriers.

## Materials and methods

### Ethics statement

Primary human tracheobronchial epithelial cells were isolated from patients (Cantonal Research Ethics Commission approval KEK-BE 1571/2019), who gave written informed consent.

### Cells and viruses

Well-differentiated human airway epithelial cell (hAEC) cultures were prepared as previously described [44]. Briefly, cryopreserved primary human tracheobronchial epithelial cells were thawed and seeded into collagen Type I-coated flasks in bronchial epithelial growth media (BEGM). After reaching 80%, confluence cells were harvested and seeded into collagen-type IV-coated porous inserts (6.5 mm radius insert, Costar, Corning, Glendale, AZ) in 24-well cluster plates. After seeding, cells are grown at a liquid-liquid interface for 2–3 days in BEGM, followed by air-liquid interface (ALI) exposure by air-lifting the cultures and changing the basolateral medium every two days to allow for cellular differentiation over a period of at least four weeks.

The human megakaryoblastoid cells UT7/Epo (kindly provided by E. Morita, Tohoku University School of Medicine, Japan) and the canine MDCK II cells were cultured in Eagle's minimal essential medium (MEM) (Sigma, St. Louis, MO) containing 5% fetal calf serum (FCS). UT7/Epo culture media was complemented with 2 U/ml recombinant erythropoietin (Epo).

ExpiSf9 cells were cultured at 27˚C and 125 rpm in ExpiSf CD Medium (Thermo Fisher Scientific, Waltham, MA). Native B19V devoid of virus-specific antibodies was obtained from infected plasma samples (CSL Behring AG, Charlotte, NC). The virus was pelleted by ultracentrifugation through a 20% sucrose cushion and further purified by iodixanol density gradient ultracentrifugation, as previously described [59]. Virus concentration was determined by qPCR. *Pseudomonas aeruginosa* and *Pseudomonas aeruginosa* bacteriophage PP7 were obtained from ATCC; 15692 and 15692-B4, respectively. PP7 production, purification, and quantification were performed as described elsewhere [14].

## Antibodies

The human monoclonal antibodies (mAb) 860-55D against intact capsids, and 1418 against the VP1u, were purchased from Mikrogen (Neuried, Germany). A polyclonal chicken anti-globoside IgY antibody was a gift from J. Müthing (University of Münster, Germany). Polyclonal rabbit antibodies against the tight junction proteins zonula occludens 1 and claudin 1 were purchased from Proteintech (Planegg-Martinsried, Germany). Alexa Fluor 647-conjugated anti-β-tubulin (9F3) rabbit mAb (Thermo Fisher Scientific) to visualize cilia. A polyclonal rabbit antibody against GAPDH was obtained from Abcam (Cambridge, UK). Recombinant VP1u was detected by immunofluorescence with a rat anti-FLAG mAb (Agilent Technologies, Santa Clara, CA), and a goat anti-Rat, Alexa Fluor 594 conjugate (Abcam).

## Generation of B3GalNT1 knockout MDCK II cell line

MDCK II cells were transfected with a plasmid to generate a double-strand break in the target gene (β-1,3-Gal-T3 CRISPR/Cas9 KO GFP), and a plasmid for homology-directed repair (β-1,3-Gal-T3 HDR RFP) (Santa Cruz Biotechnology, TX). For transfection, Lipofectamine 3000 (Invitrogen, CA) was used according to the manufacturer's instructions. Doubly transfected cells were sorted by FACS (ARIA, BD Biosciences, NJ). Two serial bulk sorting were performed to isolate and concentrate RFP-expressing cells, followed by a final single-cell sort. To verify the knockout of the B3GalNT1 gene, mRNA was isolated using the Dynabeads mRNA DIRECT Kit (Invitrogen, CA) according to the manufacturer's instructions. The Luna Universal One-Step RT-qPCR Kit (New England Biolabs, Ipswich, MA) was used. Forward primer: 5′- CCTGAGTTTCCTTGTGATGTGG -3′; reverse: 5′-CATGATGTACTTGG CATTGGGG -3′.

## Production and purification of VP2 virus-like particles

Recombinant B19 virus-like particles (VLPs) consisting of VP2 were produced using the ExpiSf Expression System Starter Kit (Thermo Fisher Scientific) following the manufacturer's instructions. Briefly, the VP2 gene was cloned into a pFastBac1 plasmid. Recombinant baculoviruses were generated by transfection of ExpiSf9 cells with the recombinant bacmids and used to infect ExpiSf9 cells at an MOI of 5. Infected ExpiSf9 cells were lysed 72 hpi by three freeze and thaw cycles with TNTM buffer (2 mM $MgCl_2$, 25 mM Tris-HCl pH 8, 100 mM NaCl, 0.2% Triton-X-100). To avoid proteolytic activity, the buffer was expanded with a cOmplete EDTA-free protease inhibitor tablet (Sigma). The VP2 particles were purified and concentrated by ultracentrifugation through a 20% (w/v) sucrose cushion in TNET (1 mM EDTA, 50 mM Tris-HCl pH 8, 100 mM NaCl, 0.2% Triton X-100) followed by a density separation on an OptiPrep (Sigma) gradient using an SW41 Ti rotor (Beckman Coulter, Indianapolis, IN). Fractions containing VLPs were identified by dot blot with a monoclonal mouse antibody against viral proteins (3113-81C; US Biological, Boston, MA) and exchanged into a storage buffer (20 mM Tris-HCl, [pH 7.8], 10 mM NaCl, 2 mM $MgCl_2$) using the PD-10 desalting columns (GE

Healthcare, Piscataway, NJ). Quantification of VP2 particles was determined by absorbance at A280 with NanoDrop (NanoDrop2000, Thermo Fisher Scientific) and by comparison to a reference BSA sample on SDS-PAGE, stained with Imperial Protein Stain (Thermo Fisher Scientific). The integrity and monodispersity of VLPs were examined by electron microscopy.

## Immunofluorescence

To detect the VP1u receptor, hAECs and MDCK II cells were incubated with recombinant VP1u at 4˚C for 1h in the presence of an anti-FLAG antibody and washed prior to fixation. The cells were fixed with methanol:acetone (1:1) at -20˚C for 4 min and blocked in 10% goat serum/PBS for 20 min. For the detection of B19V, cilia, and tight junction proteins, cells were fixed and blocked, as mentioned above, prior to staining with primary antibodies. Primary antibodies were detected with secondary antibodies conjugated with Alexa Fluor dyes (Invitrogen). For detection of globoside, the cells were fixed with 4% formaldehyde for 10 min, washed twice, quenched with 1 M Tris-HCl (pH 8.5) for 10 min, and permeabilized with 0.2% Triton X-100 in PBS for 10 min, blocked as above, incubated with anti-globoside antibody and secondary antibodies as mentioned above. The samples were washed once in milli-Q water, let dry, and mounted with a DAPI-containing ProLong Diamond mounting medium (Thermo Fisher Scientific). The samples were visualized using laser scanning confocal microscopy (Zeiss LSM 880) with a 63x oil immersion objective.

For immunofluorescence analysis with polarized cells grown on porous membranes, the apical medium was removed, and the cells were fixed with 4% formaldehyde for 10 min, washed twice, quenched with 1 M Tris-HCl (pH 8.5) for 10 min, and permeabilized with 0.2% Triton X-100 in PBS for 10 min. Cells were blocked and incubated with antibodies, as described above. The membrane was removed from the insert with a clean scalpel, mounted onto a slide with DAPI-containing ProLong Diamond mounting medium (Thermo Fisher Scientific), and covered with a coverslip.

## Virus binding and internalization

B19V was incubated with cells ($10^4$ geq/cell) at 4˚C for 1h for attachment or 37˚C for internalization. Cells were washed and prepared for immunofluorescence analysis, as described above, or qPCR. For qPCR, DNA was isolated with the GenCatch Plus Genomic DNA Miniprep Kit (Epoch Life Science, Missouri City, TX) or with the DNeasy Blood & Tissue Kit (Qiagen, Hilden, Germany) according to the manufacturer's protocol. Absolute quantification was performed using the standard curve method. Quantification of the β-actin gene was used for the normalization of data in qPCR experiments.

## Transcytosis assay

hAEC cultures were prepared on Transwell inserts (Corning, Glendale, AZ) as described above. Before the transcytosis assay, the functionality of the tight junctions was tested by measuring the transepithelial electrical resistance (TEER). MDCK II cells were seeded on TC inserts (Sarstedt) with a porous membrane (0.4 μm pore diameter, $2x10^6$ pores/$cm^2$, and 12 μm thickness). The inserts were placed in a 24-well plate with medium and incubated at 37˚C with 5% CO. The functionality of the tight junctions was assessed at increasing days post-seeding with B19V ($3x10^9$) at pH 7.4, where no interaction with globoside is possible, and with bacteriophage PP7 ($3x10^{10}$), which does not interact with eukaryotic cells. PP7 RNA was quantified by RT-PCR as previously described [14].

Cells were washed once with PBS and layered with 200 μl of apical buffer consisting of PiBS pH 6.1 (20 mM piperazine-N,N′-bis[2-ethanesulfonic acid], 121 mM NaCl, 2.5 mM KCl) or

MEM. The inserts were placed in a 24-well plate with 600 μl MEM. To quantify viruses in the apical and basolateral compartments, 50 μl medium was harvested and used for DNA extraction and quantification by qPCR, as described above. Results were normalized for the differences in volume between apical and basolateral compartments.

### Infectivity assay

The infection of hAEC cultures, MDCK II, and UT7/Epo cells was examined by quantifying viral NS1 mRNA by RT-PCR. Cells were infected with B19V ($2x10^4$ geq/cell) and incubated at 37˚C. Cells were washed four times with PBS at increasing times post-infection, and RNA was extracted with GenCatch Total RNA Miniprep Kit (Epoch Life Sciences) according to the manufacturer's protocol. The extracted samples were treated with DNase I for 30 min at 37˚C, and NS1 mRNA was quantified as described elsewhere [21]. Quantification of the β-actin gene was used for the normalization of data in qRT-PCR experiments.

### Transcytosis with inhibitors of endocytosis

All drugs were purchased from Sigma. Pitstop 2 and genistein were dissolved in DMSO, whereas chlorpromazine (CPZ), methyl-β-cyclodextrin (mβCDX), and dynasore were dissolved in water. Briefly, MDCK II cells were seeded in TC inserts (Sarstedt). The cells were pre-treated with the inhibitors at pH 7.4 and incubated at 37˚C for 1 h. Next, the cells were incubated with the drugs and B19V ($3x10^9$) in PiBS (pH 6.1) at 37˚C to allow virus attachment and uptake. After 3h, the cells were washed with PBS (pH 7.4) to remove attached but not internalized viruses. B19V was detected in cells by IF with an antibody against capsids and quantified in the basolateral compartment by qPCR, as described above.

### Statistical analysis

Data analysis was performed using GraphPad Prism and presented as the mean of three independent experiments ± standard deviation (SD). Data were evaluated by Student's t-test. For all P values, significance levels are denoted as follows: ns, not significant; *P < 0.05, **P < 0.01, and ***P < 0.001. P value less than 0.05 was considered statistically significant.

### Supporting information

**S1 Fig. VP1u constructs used for the detection of VP1u receptor expression.** (A) Schematic representation of functional (FL-VP1u, Full-length) and non-functional (N29, N-terminal truncated) recombinant VP1u constructs. RBD, receptor-binding domain. PLA2, phospholipase $A_2$. MAT, metal affinity tag (B) Detection of the VP1u receptor in UT7/Epo cells. Cells were incubated with recombinant VP1u constructs and an anti-FLAG antibody, washed, fixed, and visualized by confocal microscopy. DAPI was used to visualize nuclei.
(TIF)

**S2 Fig. B3GalNT1 knockout.** MDCK II cells were transfected with either control or B3GalNT1 KO and HDR plasmids and analysed by FACS. Cells containing both CRISPR/Cas9 KO and HDR plasmids (Q2) showed green fluorescence protein (GFP), and red fluorescence protein (RFP) expression, respectively. These cells were sorted and cultured for one month. Single-cell sorting was performed to obtain clones with stably integrated HDR cassettes expressing RFP. Successful knockout of B3GalNT1 was verified by RT-qPCR and immunofluorescence staining with an antibody against globoside.
(TIF)

**S3 Fig. Formation of functional tight junctions.** (A) MDCK II cells were seeded in TC inserts and allowed to form a polarized monolayer. To evaluate paracellular transfer, B19V ($3 \times 10^9$) was added to the apical medium at neutral pH at progressive days post-seeding. At the indicated hours post-infection, viruses were quantified in the basolateral medium by qPCR. (B) At 3 days post-seeding, the formation of tight junctions was visualized with an antibody against ZO-1 and a secondary Alexa Fluor 594. DAPI was used to visualize nuclei.
(TIF)

**S4 Fig. Transcytosis assay with two different porous membranes.** Membranes with varaying pore densities and thicknesses were tested in parallel. B19V was added to the apical side of the MDCK II cells at acidic pH (6.1), and the accumulation of viruses in the basolateral medium was quantified at increasing times post-inoculation by qPCR.
(TIF)

**S5 Fig. Z-stack imaging of MDCK II cells grown in inserts and incubated with B19V or VLPs.** Polarized MDCK II cells were grown on TC inserts for 3 days to allow the formation of tight junctions. B19V or VLPs ($3 \times 10^9$) were added at 37˚C for 1h at pH 6.1. The cells were washed, fixed, and stained with an antibody against intact capsids (green). Nuclei were stained with DAPI (blue). Z-stacks were acquired by confocal microscopy from the top of the epithelial layer to the beginning of the porous membrane (dotted line). (A) Z-projections (XY). (B) Orthogonal view (XZ). Scale bar: 10 μm.
(TIF)

**S6 Fig. Integrity of transcytosed B19V.** Following virus transcytosis, the basolateral medium was treated with micrococcal nuclease alone or in combination with NP40 prior to DNA extraction and qPCR. Basolateral media (100 μl) was incubated with 11 μl of 10X nuclease buffer (500 mM Tris-HCl pH 8.0, 150 mM CaCl$_2$), 1.1 μl of 10 mg/ml BSA, and 0.2 μl micrococcal nuclease (400 units, NEB) at 37˚C for 1h. The reaction was quenched with 11 μl 0.2 M EDTA. Alternatively, nuclease treatment was performed in combination with 0.1% NP40. As a control, the samples were heated to 85˚C for 5 minutes prior to nuclease digestion to expose the viral DNA. UT, untreated.
(TIF)

**S7 Fig. Purity and integrity of VP2 VLPs.** (A) The purity and concentration of VLPs (VP2-only particles) were examined by SDS-PAGE. Protein concentration was determined using serial dilutions of purified BSA. (B) The integrity and monodispersity of VLPs were analyzed by electron microscopy. Scale bar; 100 μm.
(TIF)

**S8 Fig. Schematic representation of the establishment of human tracheobronchial airway epithelial cell (hAEC) culture.** Primary epithelial cells are isolated from tissue biopsies, from which connective tissue is removed, using protease treatment, and grown in BEGM media to expand the cells as a monolayer. Expanded cells are seeded onto Transwell membranes and grown under the submerged condition until they reach confluence. The differentiation phase is initiated by air-lifting the cells to establish the air-liquid interface (ALI). During the 4 weeks post-ALI exposure, the AEC cultures will differentiate into a pseudostratified layer of differentiated AEC cultures, showing the phenotype of ciliated, goblet, and basal cells. Created with BioRender.com.
(TIF)

**S9 Fig. Transepithelial electrical resistance (TEER) measurement in hAECs grown in Transwell inserts.** The apical side of hAECs was exposed to different pH conditions and

infected with B19V at 37°C. At increasing post-infection times, TEER was measured manually using an Epithelial Volt/Ohm Meter (EVOM) with a "chopstick" electrode (World Precision Instruments, Hitchin, UK). A volume of 200 μl of TEER solution (4.5 g NaCl, 91.89 mg $CaCl_2$, and 1.194 g of HEPES in 500 ml of distilled water) was added to the apical chamber of the Transwells before inserting the electrodes.
(TIF)

## Acknowledgments

We thank Sabina Berezowska and Irene Ramos-Centeno (Institute of Pathology, University of Bern) for providing the tissues via the Tissue Bank Bern.

## Author Contributions

**Conceptualization:** Ronald Dijkman, Carlos Ros.

**Formal analysis:** Corinne Suter, Minela Colakovic, Jan Bieri, Mitra Gultom, Carlos Ros.

**Funding acquisition:** Carlos Ros.

**Investigation:** Corinne Suter, Minela Colakovic, Jan Bieri, Mitra Gultom.

**Methodology:** Corinne Suter, Jan Bieri, Mitra Gultom, Carlos Ros.

**Project administration:** Carlos Ros.

**Resources:** Ronald Dijkman, Carlos Ros.

**Supervision:** Ronald Dijkman, Carlos Ros.

**Validation:** Corinne Suter, Minela Colakovic, Jan Bieri, Mitra Gultom, Ronald Dijkman, Carlos Ros.

**Visualization:** Corinne Suter, Minela Colakovic, Jan Bieri, Mitra Gultom, Carlos Ros.

**Writing – original draft:** Carlos Ros.

**Writing – review & editing:** Corinne Suter, Minela Colakovic, Jan Bieri, Mitra Gultom, Ronald Dijkman.

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
