## [Decision Letter · Decision Letter 0]

2 Mar 2023

Dear Dr. Ros,

Thank you very much for submitting your manuscript "Globoside and the mucosal pH mediate parvovirus B19 entry through the epithelial barrier" (PPATHOGENS-D-23-00096) for consideration at PLOS Pathogens. As with all papers reviewed by the journal, your manuscript was reviewed by members of the editorial board and by several independent reviewers. In light of the reviews (below this email), we would like to invite the resubmission of a significantly-revised version that takes into account the reviewers' comments.

I am returning your manuscript with three reviews. The reviewers came to similar conclusions about the paper, as you will see. After reading the reviews and looking at the manuscript, I recommend Major Revision based on the critiques from the more critical reviews. I am sorry I cannot be more positive at the moment, however we are looking forward to receiving your revision. With these revisions, the manuscript will be suitable for a resubmission, if you so wish to do so. Note that we may send your paper back to some of the more critical reviewers upon resubmission.

Among the reviews, please pay particular attention to “crucial/essential” additional experiments required by Reviewer 1. You are also required to address the multiple points of clarifications raised by reviewers 2 and 3; discrepancies in figure callouts versus descriptions raised by reviewers 2 and 3; request for more robust statistical analysis from reviewer 2; and multiple editorial issues helpfully pointed out by reviewers 2 and 3.

We cannot make any decision about publication until we have seen the revised manuscript and your response to the reviewers' comments. Your revised manuscript is also likely to be sent to reviewers for further evaluation.

Sincerely,

Kinjal Majumder, PhD

Guest Editor

PLOS Pathogens

Blossom Damania

Section Editor

PLOS Pathogens

Kasturi Haldar

Editor-in-Chief

PLOS Pathogens

orcid.org/0000-0001-5065-158X

Michael Malim

Editor-in-Chief

PLOS Pathogens

orcid.org/0000-0002-7699-2064

Reviewer's Responses to Questions

**Part I - Summary**

Reviewer #1: Corinne Suter and her colleagues have elucidated the mechanism by which human parvovirus B19 is transcytosed via globoside in epithelial cells. Initially, they demonstrated that binding of B19V particles to the cell surface is essential for their internalization using MDCKII cells, and this process is VP1u-independent but dependent on low pH conditions. Additionally, they demonstrated that the binding process is reliant on both globoside and caveolae endocytosis. Moreover, they observed that this transcytosis mechanism was also present in primary human airway cells. The route of airborne transmission of B19V to target erythroid progenitor cells has not been well understood. The study has addressed a significant topic regarding the invasion of B19V virions via transcytosis of respiratory epithelial cells. Overall, the experiments were executed, and the outcomes were presented clearly. I have a few suggestions to make the message clearer and provide some useful additional information.

Reviewer #2: In their previous publications (Bieri et al. 2019, 2021) the authors demonstrated that globoside is dispensable for the B19V entry and that the acidic pH is required for the B19V attachment to globoside and this interaction is required for endocytic entry. This current paper is elucidating the role of an acidic environment for the transcytosis of B19V through epithelial model MDCK II cells and primary airway epithelial cells. These studies were performed with cells grown on porous membranes which allowed the detection of B19V entry from the apical and egress from the basolateral side. The absence of produced NS1 verified that the virus did not replicate in the cells during transcytosis. The characterization of the entry pathway suggested that B19V entry in acidic conditions is mediated by a caveolar-like entry pathway.

This is a compact, clearly written, and rather jargon-free paper. The performed experiments are well thought out, and they proceed logically answering the questions arising. However, there are major issues with the figures, as some of the figures do not correspond to the text, and figure legends, all in all complicating the revision of the reported results.

After modification, this publication is most likely highly useful for others wishing to study the regulation of viral particle transcytosis in polarized cells. Therefore, I believe this study makes a valuable contribution to the field of virus-cell interactions and how pathogenic viruses move across the barrier formed by epithelial cells using transcytosis.

Reviewer #3: B19V is a human pathogenic virus transmitted via respiratory route, although targeting erythroid progenitors in the bone marrow to achieve a productive infection. Portal and mode of entry have remained elusive so far, transcytosis being a likely mechanism. In their work, Sutter et al. present an abundant set of experimental data actually showing transcytosis of B19V through epithelial cells first in MDCK II as model, then in hAEC as a relevant system. Crucial to transcytosis, interaction of virions with globoside in acidic conditions, a key parameter for achieving a productive infection in the main target cells. Overall, the submitted paper provides convincing evidence and significantly contributes to our understanding of B19V interactions with target cells and mode of transmission, confirming the expertise of the group. Hereby, I provide some comments with the aim of a better readability and understanding of the paper.

**Part II – Major Issues: Key Experiments Required for Acceptance**

Reviewer #1: 1- Line261-262: Performing a subcellular localization study through Z stack sectioning from apical to basolateral is crucial in highlighting the potential intracellular accumulation of B19V in MDCKII cells.

2- In Fig7, it is essential to include the uptake of VLPs (VP2 only) via clathrin-independent, cholesterol, and dynamin-dependent mechanisms in primary human airway cells, which excludes VP1u involvement.

3- In Fig8, it is crucial to include the uptake and transcytosis of VLPs (VP2 only) under acidic conditions to determine if the VP2 interacts with globoside to mediate the novel uptake pathway in epithelial cells. Additionally, it is imperative to provide infectivity data of the B19V recovered in the basal medium following transcytosis through cells, as presented in the current fig3d.

4- Line 352-357: As mentioned by the authors, AXL was recently identified as a VP1u partner that facilitates B19V entry into cells. It is quite important to demonstrate that this event is distinct from AXL-mediated viral entry and to confirm that these cells do not express AXL on the cell surface.

Reviewer #2: 1. In general, the clathrin-independent, caveolin-like entry pathway of capsids from the apical side during transcytosis is well described and studied here. Does this rule out the autophagosome-associated pathway?

2. There is one aspect ongoing throughout the paper that is of concern, and that is the question of how does capsid exocytosis/exit from the basolateral side happen? Why does the acidic PH outside the cells affect the exit of the virus from the basolateral side? This important aspect is not addressed and should be introduced and discussed, and even experimented on.

3. Where is the virus blocked in absence of low pH? Specifically, lines 254-264, Figure 6B: What is the localization of the intact capsids - intracellular or on the cell surface? It seems that they might be located in intracellular vesicles. In this context, the DIC image or some marker to show the plasma membrane or outline of the cell is needed to show the possible intracellular localization. Maybe the intracellular localization could be studied by using some common markers of exocytosis (e.g. CD63, markers for ESCRT Pathway).

4. Figures 3. and 5: Figure 3C. is not explained in the main text. Figure 3E mentioned in the text is missing (line 208). The explanation of Figure 3D is missing(lines 198-201). The results of Fig 5E are not explained in the text (line 251). Please explain the results in Figs 5B, 5C, and 5D more carefully (Line 245). Increase the text size of Fig 5C's x-axis legend. Does α-capsid correspond to VP2 antibody (main text line 242 and figure legend lines 629-630)?

5. In many figures, more than two samples are compared for statistically significant differences. Student's t-test is not optimal for such multiple comparisons. A better practice is to use ANOVA combined with a suitable post hoc test (for example Tukey's or Dunnett's, or maybe Games-Howell in case of unequal variances).

Reviewer #3: Introduction. p.4, li.75ff. Mention to the recently identification of AXL as a specific ligand for VP1u is appropriate in this context.

Material and Methods, p.17, li.439. UT7/Epo have different subclones used in different labs, please specify if cells used are the original clone or a derived subclone and provide appropriate source reference.

Graphs in figures 1 (C, D, E), 2 (D, E), 3 (A-D), 4 (C, D), 5 (B, C, E), 6 (A), 8 (D, E, F) and S1-4 are all heterogenous and not consistent in Y-axis scale. Some axis are in log scale, some in linear scale, sometimes origins are labeled as ‘zero’ although clearly starting from non-zero values, sometimes origins start from non-zero values, while range and intervals of values changes from graph to graph. Importantly, when not expressed as percentage, figures are meant as absolute, while a normalization to cell number or medium sample volume would be appropriate to allow better comparison. Finally, the level of statistical significance should be stated (always missing). This heterogeneity goes to paper’s disadvantage and should be amended. Moreover, although statistically significant, differences in values between experimental categories can have quite a different meaning if considered on a linear or logarithmic scale.

Results, p.6, li130-152 and related figure 1. Either in the legend (A) or in the Material section please indicate what secondary antibody is used to reveal the VP1u-anti FLAG complex. legend (C): “viral DNA was extracted and quantified by PCR” should be “DNA was extracted and viral DNA quantified by PCR”. In figure 1: comparing the neutral (pH 7.4/7.4) condition, the detected B19V DNA in graph C is not seen in the comparable conditions in graphs D and E (and also in figure 2D). Is this correct?

Results, p.7, li-153-176 and related figure 2. Speaking of B3GalNT1 gene knock-out, data on sorting might be shown as supplementary results. However, in Figure 2, knock-out of gene expression is shown by end-point RT-PCR. More correctly, genome editing (knock-out) should be shown by PCR on genomic DNA, while assessment of gene expression silencing requires quantitative RT-PCR

Results, p.8, li.194-201 and related figure 3. In the legend, please indicate the moi of added virus rather than an absolute amount. In the text, please correct reference to panels C and D (not E). Although the absolute amount of virus recovered in the basolateral side is dependent on the neutral/acidic condition at the apical side, it is actually not clear whether this difference is also present in the relative amount of virus in basolateral side compared to virus bound by cells. In other words, does the pH affects uptake or transcytosis efficiency? Finally, infectivity of relased virus as shown by NS mRNA should also be normalized to the amount of viral DNA in infected UT7/Epo cells.

Results, p.11, li.260-264 and related figure 6B. I am not sure I could appreciate the difference between the two conditions, as claimed y authors. Please explain better…

Results, p.12, li.303. Figure S8 is actually Figure S6

Results, p.13, li.325. Data from TEER should be shown at least as supplemental figure

Discussion, p.15, li.372-374. Do authors hypothesize a role for the PLA2 activity associated to VP1u, especially for endosomal escape (see also p.9, li.231)

Discussion, p.15, li.380-382. The mentioned intracellular conditions required for productive infection are effective in the nucleus of infected (erythroid progenitor) cells, while in the present experiments it seems that virus does not enter nucleus of epithelial cells, and the observation does not seem relevant.

Discussion, p.16, li. 400-402. Can authors comment on this statement comapared to what shown in Bonsch et al, J.Virol 2008, 82:11784-91.

Discussion, p.16, li.404-407. The statement is only partly justified by the present experiments, and in any case absence of globoside would block not only transcytosis but also as amplification and presentation to the immune system following a productive infection.

**Part III – Minor Issues: Editorial and Data Presentation Modifications**

Reviewer #1: NA

Reviewer #2: 1. Figs the statistical significance P values should be explained either in the Figure legends or in the materials and methods e.g. For all P values, significance levels are denoted as follows: *P < 0.05, **P < 0.01, and ***P < 0.001.

2. The Fig 1A. Please, add an explanation of what is shown in the images.

3. Line 137. Please explain when first mentioned what term “a VP1u cognate receptor means”.

4. Line 159, 161: Explain the globoside knockout MDCK = KO MDCK when first mentioned. In the figure legend 2A “and transfected cells (KO)” should be “globoside KO transfected cells”Lines 167-170 sounds like materials and methods.

5. Line 170: While B19V DNA was …

6. Figure legend S2A. Could you please explain what is ZO since it is used first time here.

7. Line 205. It might be good to explain that when treated at 85°C the capsids were most likely disintegrated and their genome was exposed to nuclease.

8. Line 215: Explain TC - tricellular contact?

9. Line 253-254, 264-265, 295-296: add space

10. Fig 7B. Add the units of the used chemical concentrations to the x-axis legends since they are not explained elsewhere. How can transcytosis of the virus be more than 100% in some of the treatments? If this is due to the experimental set-up, please explain it more carefully in the results section.

11. Line 303: Should Figure S8 be S6?

12. Line 310: What does the cell body mean? What about just cells? Line 358 etc.:

13. The figures should not be mentioned in the Discussion.

14. Supplementary figure legends: Please mention the cells used.

Reviewer #3: (No Response)

PLOS authors have the option to publish the peer review history of their article (what does this mean?). If published, this will include your full peer review and any attached files.

Reviewer #1: No

Reviewer #2: No

Reviewer #3: No
---

## [Decision Letter · Decision Letter 1]

1 May 2023

Dear Dr. Ros,

Thank you very much for submitting your manuscript "Globoside and the mucosal pH mediate parvovirus B19 entry through the epithelial barrier " (PPATHOGENS-D-23-00096R1) for review by PLOS Pathogens. Your manuscript was fully evaluated at the editorial level and by independent peer reviewers. The reviewers appreciated the attention to an important topic. Based on the reviews, we are likely to accept this manuscript for publication, providing that you modify the manuscript according to the review recommendations.

I am returning your manuscript with all three reviews. All reviewers were appreciative of the value of the research. As you can see, further experiments are not necessary for this manuscript to meet the criteria for publication at PLOS Pathogens. Two of the three reviewers were satisfied with the submitted revisions and reviewer 3 had some minor edits that need to be addressed (see below). We therefore ask you to modify the manuscript according to reviewer 3’s recommendations before we can consider your manuscript for acceptance. The remaining minor revisions, raised by reviewer 3, with regards to graphical presentation and normalization of data, must be addressed to prepare the manuscript for publication. If met, I hope to be able to make a final decision without sending the manuscript out for another round of review.

Sincerely,

Kinjal Majumder, PhD

Guest Editor

PLOS Pathogens

Blossom Damania

Section Editor

PLOS Pathogens

Kasturi Haldar

Editor-in-Chief

PLOS Pathogens

orcid.org/0000-0001-5065-158X

Michael Malim

Editor-in-Chief

PLOS Pathogens

orcid.org/0000-0002-7699-2064

Reviewer Comments (if any, and for reference):

Reviewer's Responses to Questions

**Part I - Summary**

Reviewer #1: The authors appear to have addressed all of the raised concerns, and as a result, the manuscript now appears suitable for publication in PLoS Pathogens.

Reviewer #2: The authors have responded to all the reviewer comments and revised the manuscript accordingly.

Reviewer #3: I confirm my overall appreciation of the submitted manuscript, the paper is of high value, and a source for future research in the field. On general terms, considering the actual revision, my opinion is that authors could have been more compliant to reviewers’ suggestions. I acknowledge that almost all of my own suggestions have been replied in the appropriate way, nonetheless still I would ask for amendments in a few instances, critically concerning graphs.

**Part II – Major Issues: Key Experiments Required for Acceptance**

Reviewer #1: (No Response)

Reviewer #2: No additional experiments are required.

Reviewer #3: None

**Part III – Minor Issues: Editorial and Data Presentation Modifications**

Reviewer #1: (No Response)

Reviewer #2: No additional comments.

Reviewer #3: Concerning data presentation in graphs, although normalization is now mentioned by authors in the methods section and figure captions, most figures still report quantities on Y-axis as absolute numbers. Except for % values, these still need to be expressed as normalized to some other reference quantity (e.g., number of cells, viral genome copies, etc…).

PLOS authors have the option to publish the peer review history of their article (what does this mean?). If published, this will include your full peer review and any attached files.

Reviewer #1: No

Reviewer #2: No

Reviewer #3: No

Figure Files:

Data Requirements:

Reproducibility:

References:

---

## [Editor Report · Decision Letter 2]

3 May 2023

Dear Dr. Ros,

We are pleased to inform you that your manuscript 'Globoside and the mucosal pH mediate parvovirus B19 entry through the epithelial barrier' has been provisionally accepted for publication in PLOS Pathogens.

Best regards,

Kinjal Majumder, PhD

Guest Editor

PLOS Pathogens

Blossom Damania

Section Editor

PLOS Pathogens

Kasturi Haldar

Editor-in-Chief

PLOS Pathogens

orcid.org/0000-0001-5065-158X

Michael Malim

Editor-in-Chief

PLOS Pathogens

orcid.org/0000-0002-7699-2064
---

## [Editor Report · Acceptance letter]

19 May 2023

Dear Dr. Ros,

We are delighted to inform you that your manuscript, "Globoside and the mucosal pH mediate parvovirus B19 entry through the epithelial barrier," has been formally accepted for publication in PLOS Pathogens.

Best regards,

Kasturi Haldar

Editor-in-Chief

PLOS Pathogens

orcid.org/0000-0001-5065-158X

Michael Malim

Editor-in-Chief

PLOS Pathogens

orcid.org/0000-0002-7699-2064